# The Estimation of the Potential for Using Smart-Trackers as a Part of a Medical Indoor-Positioning System

Irina V. Pospelova [1,2], Irina V. Cherepanova [1,2,*], Dmitry S. Bragin [2], Ivan A. Sidorov [3], Evgeny Y. Kostyuchenko [2] and Victoriya N. Serebryakova [1]

[1] Cardiology Research Institute, Tomsk National Research Medical Center, Russian Academy of Sciences, Kievskaya Str. 111A, 634012 Tomsk, Russia; piv@csp.tusur.ru (I.V.P.); vn@cardio-tomsk.ru (V.N.S.)

[2] Faculty of Security, Tomsk State University of Control Systems and Radioelectronics, 634050 Tomsk, Russia; bds@csp.tusur.ru (D.S.B.); key@fb.tusur.ru (E.Y.K.)

[3] Irkutsk Supercomputer Center of SB RAS, 134, Lermontova, 664033 Irkutsk, Russia; ivan.sidorov@icc.ru

\* Correspondence: siv@csp.tusur.ru; Tel.: +7-913-881-77-33

**Abstract:** This research aims to estimate the feasibility of using smart-bracelets as a part of a medicine indoor-positioning system, to monitor the health status and location of patients in a hospital. The smart-bracelet takes on the role of a token of the system and can measure pulse, blood pressure and saturation and provide data transmission over the BLE. The distance between token and anchor was calculated by the RSSI. The position of a token and anchor relative to each other was determined by the trilateration method. The results of the research showed that the accuracy of the developed system in a static position is 1.46 m and exceeds 3 m in a dynamic position. Results of experiments showed that measurements from the smart bracelets are transmitted to the server of the system without distortion. The study results indicated that smart-bracelets could be used to locate patients inside a hospital or estimate their current health state. Given the low accuracy of systolic pressure measurement, it is recommended to develop an algorithm that will allow smooth measuring error for higher-precision estimation of the patient's general health state. In addition, it is planned to improve the positioning algorithm.

**Keywords:** telemedicine; indoor-positioning system; remote health monitoring; Bluetooth Low Energy; smart-bracelet

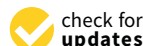



## 1. Introduction

Scientific and technological advances aim to increase innovations in the prevention, diagnosis, treatment and rehabilitation of patients with cardiovascular morbidity. One of the fields of medicine is remote health state monitoring of the patient inside a medical center. Data processing intellectualization with remote monitoring technologies gets information about the patient's state faster. Therefore, it takes on necessary measures in a timely manner. This area is named "Digital medicine", and it pertains to the foreground task of healthcare. The main aim of the integrated digital technologies is an improvement of efficiency and quality of patient's health services involved with geolocation of objects inside medical centers in real-time. Satellite navigation facilities like GLONASS, GPS, etc., are inappropriate for this problem-solving due to several factors. Walls, inter-floor construction and electromagnetic interference caused by electrical appliances operating dramatically reduce positioning accuracy or promote total satellite signal loss in-building. In this regard, there is a need to develop novel, more robust ways of allowing the tracking of object position within a closed space [1–6]. Indoor positioning can be used in many fields of human activity, including the telemedicine sphere. Combining geolocation with health monitoring systems (e.g., monitoring of pulse, blood pressure, saturation, etc.) is a promising direction of telemedicine technology development; particularly in topical use of such systems to continuously remotely monitor patients who received a surgery (for

example, a coronary artery bypass surgery). Most often, patients are able to ambulate by themselves around the hospital after the conduct of such surgeries. It is important to locate patients timely when they have post-surgery complications and rapid deterioration in their health indicators for medical emergency response. Therefore, the indoor-positioning systems (IPS) capable of health monitoring will promote the patients' mortality rate if they integrate into medical facilities.

A range of IPS uses in the medical field have been reviewed in the research. Most of them are based on radiofrequency positioning technologies (more specifically, NFC, LoRaWAN, RFID, UWB, WiFi and BLE) [7–37]. A part of reviewed systems combines several technologies at once [7,12–16,20,34]. Furthermore, the medical IPS based on passive infrared technology [38] and inertial sensors [39,40] can be distinguished.

The system, described by a group of foreign colleagues, can track people and equipment locations in real-time inside medical buildings through BLE technology. The authors argue that sensors tracking the health indicators of a patient can be integrated into this system [7]. At the same time, there are no adduced experimental investigations on the current patient health state tracking combined with this current location tracking.

The work represented by Mouhammad et al. [8] describes the development of the infection control system based on BLE technology. The described system intended to locate health professionals and track the execution of disinfection by himself after contact with a patient.

The work published by Mohsin et al. [9] describes the IPS based on BLE. The IPS is destined for tracing patients' ambulation in a hospital building. This system allows for an estimation of the patient's motion activity after surgical treatment.

The system represented by Frisby et al. [10] was developed based on BLE. This system is intended for accident and emergency departments. The system determines the healthcare delivery to a patient admitted to the emergency department. A doctor's location was determined by the room level as part of the described system. In other words, the positioning result is calculated in a rough way rather than with defined accuracy. The system registered the fact of healthcare delivery in the cases when a doctor was standing in a wireless range of an anchor for a long time period.

Power L., Jackson L. иDunnett S. represent in their work [11] a system based on BLE. This system allows the medical staff remote monitoring of the activities of daily living of the patients who have home treatment. The system makes up the individual patterns of the activities of daily living of patients during its operation. The system alarms the medical staff in the case of disturbance of one or another pattern.

Trigo J. D. et al. describe the development of the IPS based on LoRaWAN and NFC in their work [12]. The system allows keeping track of the ambulation of patients and equipment from the intensive care unit within the perimeter of a hospital consist of several buildings associated among themselves by a tunnel system. The developed system promotes coordination among employees from different departments when they operate with the same patients and equipment.

The paper represented by Yoo et al. [13] describes the development of IPS based on BLE and WiFi. The system is intended for keeping track of medical equipment locations. Authors argue that the system keeps track of equipment location over WiFi in real-time, but the polling interval of tokens over the BLE is one time per several hours. This has been done for battery saving, and it is the cause of the loss of positioning accuracy of the system.

IPS described in a paper by Yamashita et al. [14] determines the location of patients and medical staff with a combination of WiFi, BLE and geomagnetism technologies.

The work represented by Hayati and Suryanegara [15] described a positioning system for patients with psychiatric disturbance based on LoRaWan. The main function of the system is remote monitoring of patients who exit the medical staff observation zone. This reduces the risk of becoming mischances or suicides.

Hong et al., in their paper [16], describe the IPS based on UWB. This system continuously monitors medical staff inside a hospital. Moreover, the system estimates the workload of staff to distribute it more evenly

Luo et al., in their work, describe the development of the IPS based on passive infrared radiation [38]. The system is intended for remote monitoring of patients who have home treatment (e.g., elderly people). The main advantage of this system is the lack of tokens. Pyroelectric infrared sensors take on the role of anchors of the system. The sensors can locate a human by the heat radiated from their body. The system can differentiate types of patient activity apart from localizing them. Among other things, the IPS can determine when a patient falls and lies without movement for a prolonged period. In such cases, the system can alarm the operator of the IPS or first responders. The main disadvantage of the passive infrared technology is in the disability of patient identification when several people live in the same room at one time.

Works [19,20,39] describe IPS based on RFID for monitoring elderly people at home. It is worthwhile noting that the IPS described in [39] also combines inertial sensors (like gyroscope and accelerometer). Both reviewed systems detect only the patient's position but leaves out vital signs monitoring.

Koppar et al., in their work [40], describe the development of the IPS based on inertial technology. The system can be used in healthcare. The authors developed a wearable device that includes several inertial sensors and locates users of the system.

Systems described in works [21–23] are based on BLE. The user's smartphone is used as a wearable positioning device. The main distinction of these systems from the above-reviewed allows them to be suitable both for locating patients by medical staff and for patients' navigation inside the hospital.

Works [24–27] describe IPS based on WiFi. Article [25] represent the IPS, combining WiFi and BLE. The system allows tracking older people in their houses. In most cases, systems based on WiFi use smartphones, but the IPS represented in [26] uses smart-watches, and the system described in [27] can use smart-bracelets.

The system described in [28] is based on smart-bracelets maintaining BLE and estimates the current health status of a user. However, the IPS can not monitor a current user's location.

The IPS described in works [29–31] are based on BLE and allow tracking the current position of patients. Smart bracelets function as wearable devices of the system, but authors of paper [30] argue that their IPS can be used as both a smart bracelet and a smartphone.

Systems described in [32–38] are based on BLE and allow remote monitoring of a user's vital signs but can not locate their current position.

Most of the reviewed systems aim to locate objects, but there is no information about their ability to monitor vital signs remotely. Furthermore, most of the reviewed systems use their own solutions as wearable hardware, which is not universal. Using ready smart bracelets as wearable hardware as a part of the medical IPS is a more appropriate solution because there is a broad choice of this kind of device on the market. Therefore, it is possible to choose the most relevant device in a price-performance ratio. Such devices can synchronously measure several parameters of health (for example, pulse, blood pressure, saturation, etc.) and transmit them to a wireless communication channel. Moreover, it is possible to measure the received signal strength indicator (RSSI) and determine the current location of the smart bracelet. In view of this, using smart bracelets as a part of the medical IPS allows estimating both the common patient's health status and locating him. In the context of exploitation, smart bracelets do not constrain the movements of patients or cause discomfort.

This paper describes the development of an IPS based on BLE. The main function of this system is in remote monitoring, including keeping track of the current patient's location and their current health state. A wearable smart-bracelet, allowing to measure pulse, blood pressure and saturation, is used as a token of the system. The research aims to investigate the ability to use wearable smart-devices as a part of medical IPS. The system

generates alarms to the operator due to the analysis of patients' health indicators when critical situations occur.

## 2. Materials and Methods

### 2.1. Architecture of the IPS

BLE technology is used as the positioning technology for the IPS described in this paper. The architecture of the IPS is represented in Figure 1.

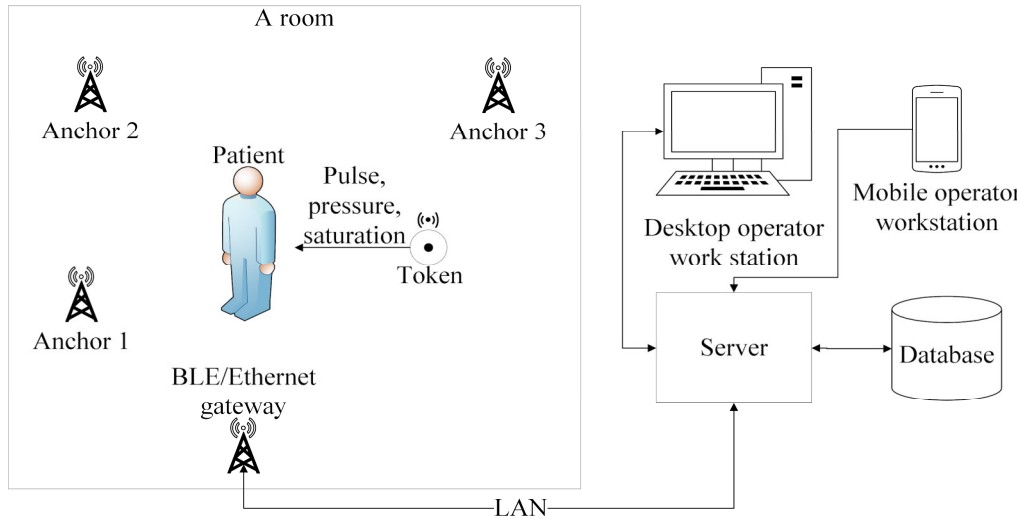

**Figure 1.** The architecture of the IPS.

As can be seen from Figure 1, the IPS includes the following parts:

- Anchors are represented as immobile BLE base stations. Anchors are mounted along the perimeter of the building, and their coordinates are known to the IP in advance.
- Tokens are represented as wearable devices equipping patients. Tokens perform two main functions. The first function is the measuring and transmission of health indicators of the patient (pulse, blood pressure and saturation). The second function is the measuring and transmission of RSSI (received signal strength indicators) from all anchors that are in the range of communication. RSSI are used for the calculation of the current position of a token.
- BLE/ethernet gateways combine wireless and wired networks, in doing so, allowing data transmission from anchors to the IPS server.
- An IPS server receives information about patients' health indicators and measured RSSI levels. The server calculates the token's current position and the patient's current health state based on the received information.
- A database contains information for correct IPS functioning.
- An operator work station (OWS) is intended for the data processing results visualization are received from the IPS server. The system described in this paper provides for two types of OWS: desktop and mobile.

### 2.2. Choosing Measuring and Communication Equipment

It was resolved to use smart-bracelet WR41 from the GSMIN producer. The intellectual property rights of the producer GSMIN were not violated during the IPS development. It was resolved to use multi-protocol microcircuit nRF52840 from the Nordic Semiconductor company for the development of BLE anchors. This choice is caused by the need to provide compatibility between anchors and smart-bracelets. According to producer information [41], these microcircuits are intended for operating with wearable devices (such as smart-watches, contactless payment devices, etc.) and for health monitoring and internet of things solutions.

### 2.3. Estimation of Measuring Accuracy and Measuring Rate of the Smart-Bracelet

A series of measurements on different test subjects were made to estimate the accuracy of health indicator registrations made by a smart-bracelet. In total it there was within the order of 50 measurements of pulse, blood pressure and saturation. Each of the obtained measurements was compared with the pulse, blood pressure and saturation values of research subjects measured by a medical tonometer and pulse oximeter.

A series of measurements on different test subjects with and without allowances made for physical exercises were made to estimate the capability to determine by the smart-bracelet abrupt changes of health indicators. In the first instance, there are the patient's health indicators (pulse, blood pressure and saturation) measured in quiescence. Then when a test subject has done strenuous exercises, their health indicators were measured again.

The rate of measure and transmission of data were sent to the server of IPS to estimate the usability of the smart-bracelets for monitoring patient health status in real-time.

### 2.4. Positioning Realization

Positioning based on radiofrequency technology divides into two large stages. The first stage is the determination of a distance between token and anchors. The second stage is in the determination of a token and anchor positions relative to each other. It was resolved to use RSSI measurement for the calculation of a distance between token and anchors.

#### 2.4.1. Calculation between Token and Anchors

Calculation of the distance between token and anchors by the RSSI method includes two stages:

1.　Filtering stage. The power of the received signal is not a constant value due to the ambient noise existence, even if a positioning object is in a static position. It is necessary to use special program filters to minimize the calculated positioning error. The Kalman filter is used to process the received RSSI in this paper.
2.　Calculation of the distance using RSSI values is corrected in the previous stage.

Data receiving and data transmission through the BLE interface in the IPS described in this paper are executed over three channels: 2.402 GHz, 2.424 GHz and 2.480 GHz. Due to this, all RSSI values are grouped according to channel numbers from which they have arrived. Then, RSSI values from each channel are processed through the Kalman filter. At the end of filtration, three anchors with the highest RSSI values are chosen for each channel. The calculation of distances between tokens and corresponding anchors is executed based on this data.

#### 2.4.2. Calculation Token and Anchors Positions Relative to Each Other

Trilateration algorithm allows determining token location over three anchors. Three anchors with the highest RSSI values are chosen to estimate the token location. The calculation of token coordinates is based on geometric computations. Distances from the token to each of the three anchors are measured in the first step of the algorithm. Each distance is taken as the circle radius with the center in the area of an anchor. In this case, a token is located at the point of intersection of three circles. If coordinates of the center and radius of each circle are known, then the position of a token can be calculated as follows:

$$\begin{cases} (x - x_1)^2 + (y - y_1)^2 = r_1^2 \\ (x - x_2)^2 + (y - y_2)^2 = r_2^2 \\ (x - x_3)^2 + (y - y_3)^2 = r_3^2 \end{cases}, \tag{1}$$

where:

- $x_1, x_2, x_3, y_1, y_2, y_3$ are coordinates of anchors;
- $x, y$ are coordinates of a token;
- $r_1, r_2, r_3$ are the distances from a token to each anchor.

However, distances between tokens and anchors are calculated with inaccuracy under real-life conditions. Therefore, it is impossible to achieve accuracy during solving the equation system (1). In view of this, it is possible to approximately solve the equation system (1) using different methods of computational mathematics to achieve the required accuracy. The range of methods described in works [42,43] are used to calculate the approximate coordinates of a token in this paper. The algorithm of token position calculation is represented in Figure 2.

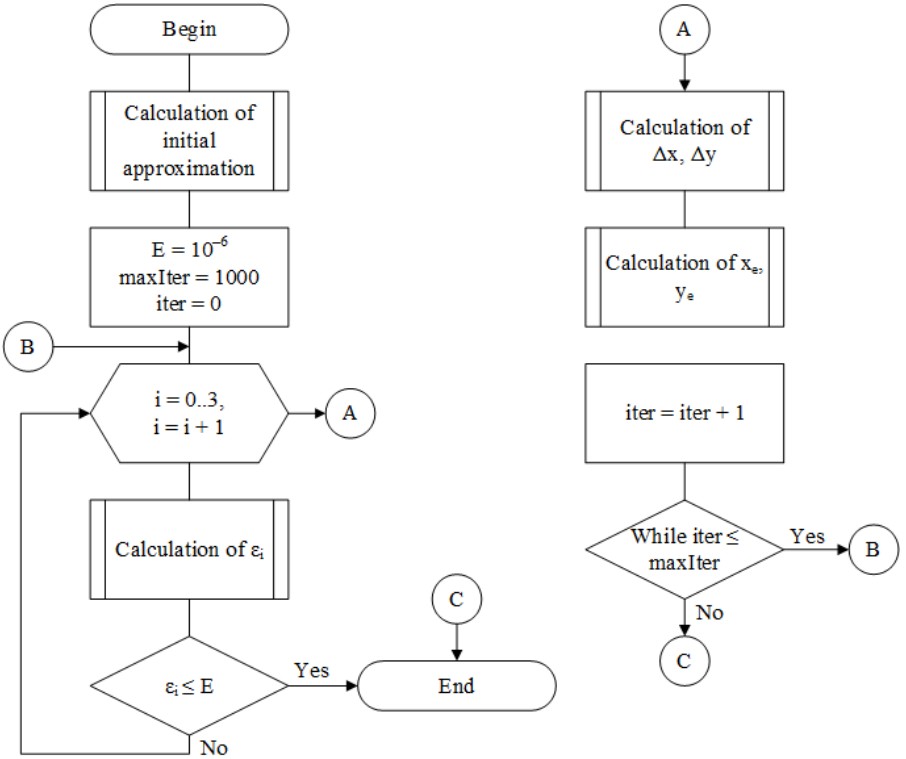

**Figure 2.** The algorithm of calculating the approximate coordinates of a token.

The equation system (1) can be represented as a matrix equation of type $A * X = B$ to calculate the initial approximation where:

$$X = \begin{pmatrix} x^2 + y^2 \\ x \\ y \end{pmatrix}, \tag{2}$$

$$A = \begin{pmatrix} 1 & -2x_1 & -2y_1 \\ 1 & -2x_2 & -2y_2 \\ 2 & -2x_3 & -2y_3 \end{pmatrix}, \tag{3}$$

$$B = \begin{pmatrix} r_1^2 - x_1^2 - y_1^2 \\ r_2^3 - x_2^2 - y_2^2 \\ r_3^2 - x_3^2 - y_3^2 \end{pmatrix}, \tag{4}$$

Roots of the equation system (1) are found with the gradient descent method after calculating the initial approximation. For this purpose, there is a need to calculate the difference between measured and computed distances as follows:

$$|\varepsilon_i| = \left| d_i - \sqrt{(x_i - x_e)^2 + (y_i - y_e)^2} \right|, \tag{5}$$

where:

- $\varepsilon_i$ is the difference between measured and computed distances of the *i*-th equation of the system (1);
- $d_i$ is a measured distance between the token and the *i*-th anchor;
- $x_i$, $y_i$ are coordinates of the *i*-th anchor;
- $x_e$, $y_e$ is an estimation of the anchors' coordinates on the current iteration.

The initial approximation, calculated on the first step of the algorithm, is used as the values of the first iteration. In following iterations, $x_e$ and $y_e$ values are calculated using Formula (6):

$$\begin{cases} x_e = x_e + step{\cdot}\Delta x \\ y_e = y_e + step{\cdot}\Delta y \end{cases}, \tag{6}$$

where:

- Step is the step value of an iteration;
- $\Delta x$, $\Delta y$ are the increases of an estimation of anchor coordinates, which can be represented as vector (7).

$$\Delta = \begin{pmatrix} \Delta x \\ \Delta y \end{pmatrix}, \tag{7}$$

In one's turn, the vector $\Delta$ can be calculated as follows:

$$\Delta = \left( B^T B \right)^{-1} B^T \varepsilon, \tag{8}$$

In such a case, the matrix $B$ is calculated as follows:

$$B = \begin{pmatrix} \dfrac{\partial \varepsilon_1}{\partial x_e} & \dfrac{\partial \varepsilon_1}{\partial y_e} \\ \dfrac{\partial \varepsilon_2}{\partial x_e} & \dfrac{\partial \varepsilon_2}{\partial y_e} \\ \dfrac{\partial \varepsilon_3}{\partial x_e} & \dfrac{\partial \varepsilon_3}{\partial y_e} \end{pmatrix}, \tag{9}$$

The calculations will be executed until $\varepsilon_i$ will approximate the required accuracy. The calculations will be executed until $\varepsilon_i$ will approximate the required accuracy equal to E, or the iterations exceed the value of the maxIter variable (Figure 2). It is recommended to use additional filtering to correct the computational results. The Kalman filter is used for this purpose in this paper.

It is necessary to increase the amount of anchors in such a way as to separate the room into coverage areas of the anchors' group when using IPS in a room with a larger floor area. At the same time, each group should include no less than three anchors to execute trilateration successfully and decrease the positioning error.

### 2.5. Monitoring the Health Indicators

As articulated earlier, the measuring equipment represented by smart-bracelets allows keeping track of such patient indicators like pulse, blood pressure and saturation. The process of monitoring the health indicators of a patient can be schematically represented in Figure 3.

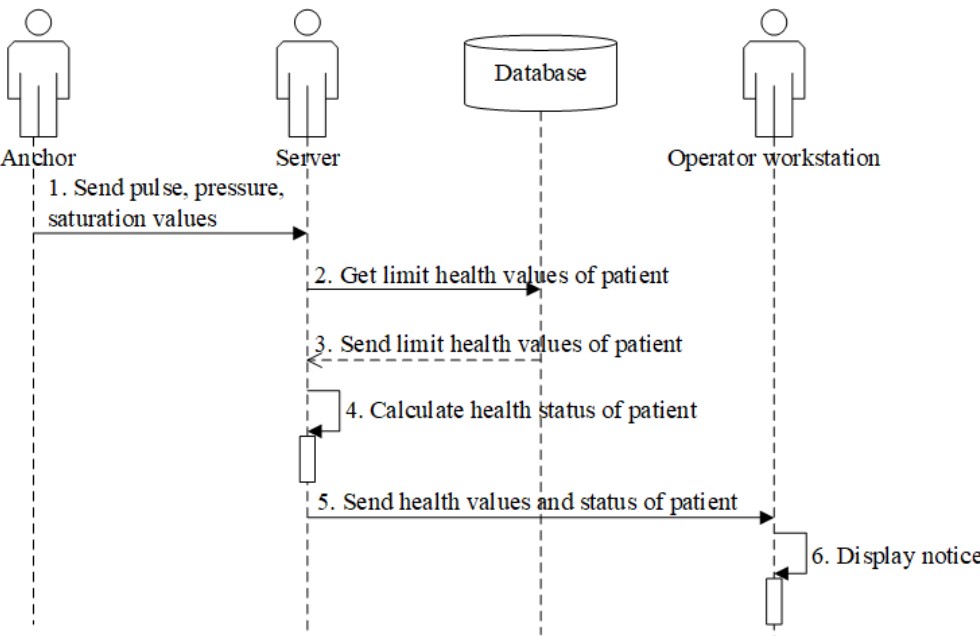

**Figure 3.** The process of monitoring the health indicators of a patient.

As can be seen in Figure 3, an anchor transmits to the server a message containing information about measured health indicators of a patient (pulse, blood pressure, saturation). Information about the limit values of one or another indicator for the specific patient is stored in the database of the IPS. Therefore, when the server receives health indicators of a patient, it sends a request to the database for receiving limit values to estimate data transmitted from a token. Then the server estimates the current patient health status and sends it to the OWS together with the measured indicators. The health status of a patient can take on one of the following values:

- NORMAL. It means that the current health indicators of a patient are within normal limits.
- WARNING. It means that the current health indicators of a patient are at the edge of the normal limits.
- FATAL. It means that the current health indicators of a patient are out of the normal limits.

If the current patient status is WARNING or FATAL, then the OWS generates a corresponding notification.

## 3. Results

The development of IPS with the function of patient health monitoring (pulse, blood pressure, saturation) is described in this paper. The system testing has been executed in a room with a size 745 × 680 cm (Figure 4).

As seen in Figure 4, three BLE anchors were mounted in a room. The coordinates of anchors were (722; 71), (702; 642) and (84; 30).

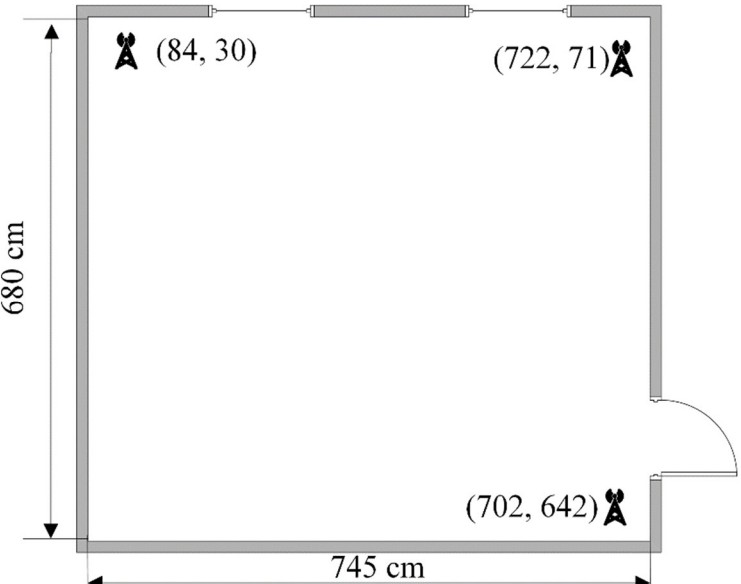

**Figure 4.** Plan of the premise for testing of the IPS.

### 3.1. Positioning

The testing of the positioning function was executed in two variants: when a token had static and dynamic positions. Values of average, maximal and minimal errors of token's coordinates computing in static are represented in Table 1.

**Table 1.** Errors of token's coordinates computing in static.

| Error on the X-Axes in cm | | | Error on the Y-Axes in cm | | |
|---|---|---|---|---|---|
| **Average** | **Maximal** | **Minimal** | **Average** | **Maximal** | **Minimal** |
| 146 | 227 | 68 | 81 | 204 | 13 |

As can be seen from Table 1, the average error on the *X*-axis is 146 cm and on the *Y*-axis is 204 cm. In total, the error on the *X*-axis did not exceed 227 cm, and on the *Y*-axis, it did not exceed 204 cm.

The result of a token positioning in dynamic is represented in Figure 5.

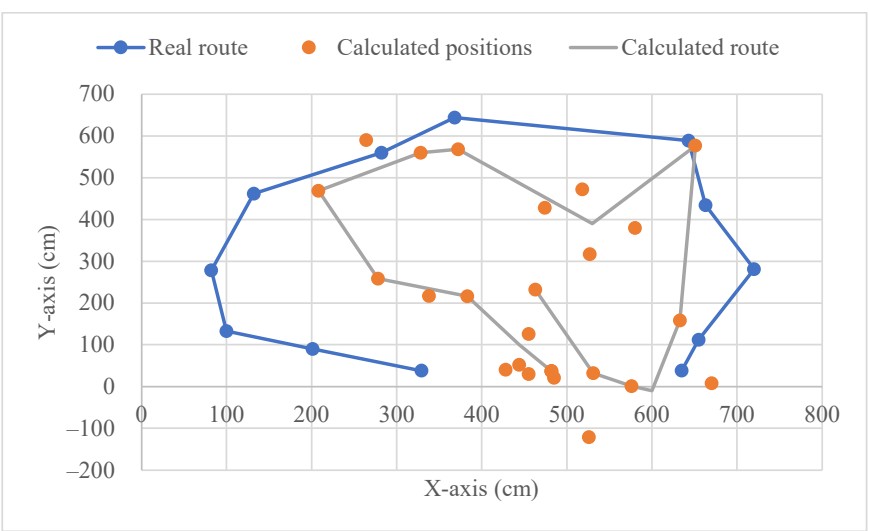

**Figure 5.** The result of a token positioning in dynamic.

As can be seen in Figure 5, the positioning accuracy of a token in dynamic is much worse than in static. In this case, the locating error was 2.5 m. The error was three or more meters in some points of a room. In view of this, it will be planned to improve the positioning accuracy with positioning and filtering algorithms modification in the future and also increase the number of anchors.

### 3.2. Estimation of Smart-Bracelet Measurements

Series of measurements on different test subjects were made to estimate the accuracy of health indicator registrations made by a smart-bracelet. In total, it was made within the order of 250 measurements of pulse, blood pressure and saturation. The analysis of experimental results revealed that the average measuring error of diastolic pressure and pulse fell within the limits of 4%. The average measuring error of saturation fell within the limits of 1%. The average measuring error of systolic pressure fell within the limits of 19% that does not correspond to claims of the smart-bracelet producer.

The experiment in the estimation of changing vital signs showed that the bracelet could register an intense rise in pulse and blood pressure and a fall in saturation level after doing physical exercises.

Using smart-bracelets as a part of IPS shows that measuring and transmitting health indicators occurred once per 45 s on average. The RSSI value between a smart-bracelet and the anchors locating it to a communication radius, to calculate token positions, are measured one time per 100 ms.

### 3.3. Visualization of the IPS Functioning

The OWS was developed for visualization of the IPS functioning. The general appearance of the OWS is represented in Figure 6.

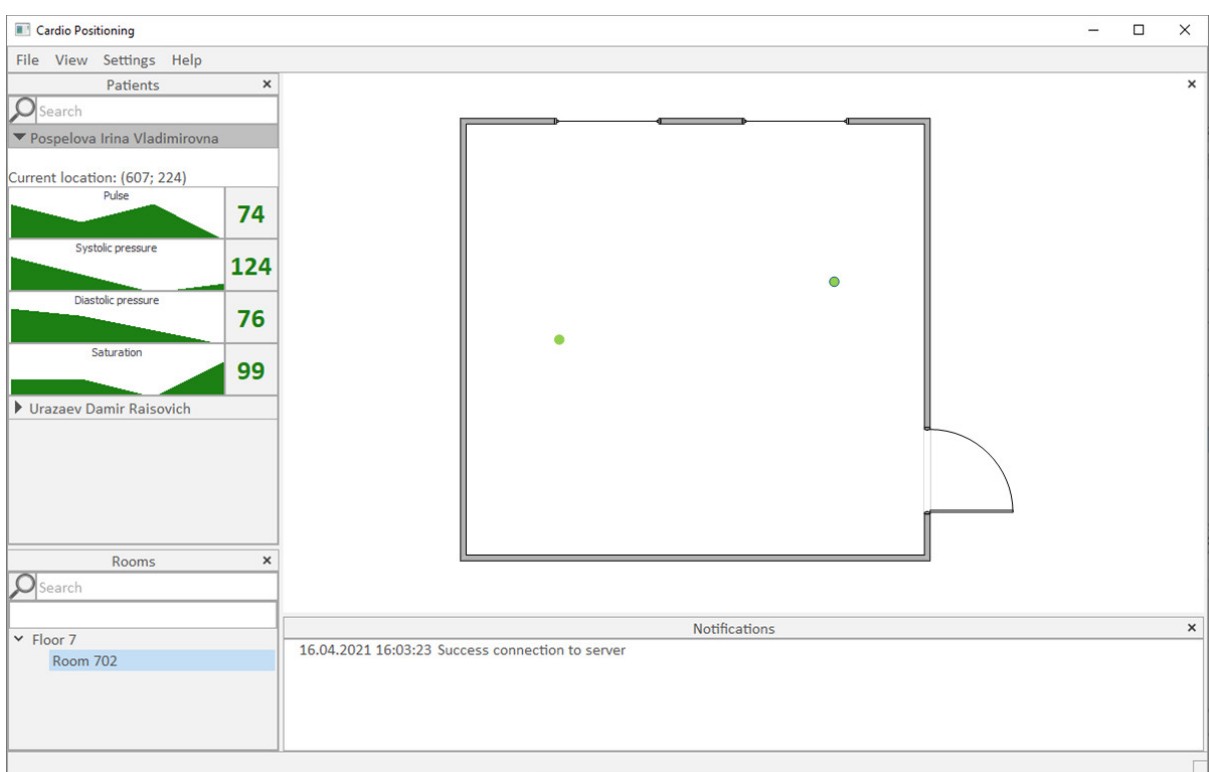

**Figure 6.** The graphical user interface of the OWS.

As can be seen in Figure 6, the graphical user interface of the OWS includes the following components:

- The panel of patients displays current coordinates and the main health indicators of each patient (pulse, saturation, systolic and diastolic pressure). The displayed health indicators have a color-coded indication to provide visibility. If health indicators are within normal limits, then they are colored green. If health indicators are at the edge of the normal limits, then they are colored yellow. Critical health indicators are colored red.
- The interactive map is represented as a 2D plan of the premises in which IPS is deployed. Patients on the map are marked with points. The color of a point depends on the current health state of a patient (green, yellow or red). The color indication is executed on the same principle as the health indication from the panel of patients.
- The panel of notifications displays the main events occurring during the IPS functioning. It also displays the time of an event occurring. All displayed events also have a color indication
- The panel of premises displays a hierarchical tree of the rooms in which the IPS is deployed.

## 4. Discussion

The accuracy of token positioning within the developed IPS was estimated in two states: static and dynamic. The average error of a token positioning in a static position is 1.46 m, but it does not exceed 2.27 m. The error of a token positioning in dynamic exceeds 3 m. Based on the research results, it can be concluded that the accuracy of positioning is insufficient at the current stage of IPS development. Given this, it will be planned to increase the number of anchors in the system and improve positioning and filtering algorithms in the future. The health indicators received by the server from smart-bracelets transmit to the IPS one time per 45 s. This allows receiving information about patient health status with a well-speed in real-time. The accuracy of measuring health indicators allows estimating the current patient state. Furthermore, smart-bracelets are able to register the abrupt change in the patient's pulse and blood pressure. According to the obtained data, it can be concluded that smart-bracelets are suitable for estimating both the current location and health indicators of a patient. It is recommended to use algorithms for estimating health states based on all measured indicators at one time (pulse, blood pressure and saturation).

The results of the IPS functioning are visualized with OWS. The OWS is equipped with a 2D plan of the premises. This plan displays the current location of all patients registered in the system. The health indicators of each patient are displayed on the special panel as diagrams with a color indication. The color of the indicators depends on the current health state of a patient. All events occurring during the IPS functioning are registered in the panel of notifications and a special system log. Moreover, the panel of notifications displays messages with the corresponding color indication when a critical situation exists.

## 5. Conclusions

The applicability of smart-bracelets as tokens of the medical IPS for health and location monitoring of patients was researched in this paper. The experimental results showed that the smart-bracelets have acceptable measuring accuracy of the vital signs to the exclusion of systolic pressure. In addition, they can register an abrupt change of vital signs of patients. The measuring rate of health indicators and time of its transmission to the IPS server allows using the smart-bracelets for patient monitoring in real-time. It is possible to find the current location of the smart-bracelets by a combination of the RSSI method, trilateration and Kalman filter because they are able to communicate over the BLE interface. It succeeded in developing medical IPS based on smart-bracelets with the capability of monitoring current values of pulse, blood pressure, saturation and location of patients remotely. According to the results of the research, it can be concluded that the smart-bracelets can be used as the tokens of the IPS both for a general estimation of the current health state of a patient and for the determination of their current location. The development of an algorithm of estimation of the general health status of a patient to allow

smoothing the poor accuracy of systolic pressure measurements is planned in the future. In addition, it is planned to improve the positioning and data filtering algorithm aimed to increase positioning accuracy. Furthermore, it is planned to increase the number of anchors in the IPS with the same aim.

**Author Contributions:** Conceptualization, D.S.B. and V.N.S.; methodology, I.V.P.; software, I.V.P. and I.V.C.; validation, D.S.B. and V.N.S.; formal analysis, I.V.P.; investigation, I.V.P. and I.V.C.; resources, D.S.B.; data curation, D.S.B. and I.A.S.; writing—original draft preparation, I.V.P., I.V.C. and E.Y.K.; writing—review and editing, D.S.B. and E.Y.K.; visualization, I.V.P., I.A.S. and I.V.C.; supervision, D.S.B.; project administration, V.N.S.; funding acquisition, E.Y.K. All authors have read and agreed to the published version of the manuscript.

**Funding:** This research was funded by the Ministry of Science and Higher Education of the Russian Federation within the framework of scientific projects carried out by teams of research laboratories of educational institutions of higher education subordinate to the Ministry of Science and Higher Education of the Russian Federation, project number FEWM-2020-0042.

**Institutional Review Board Statement:** The study was conducted in accordance with the principles of the Declaration of Helsinki, and it was approved by the Institutional Review Board. The research represented in this paper was not registered in The Clinical Trial Registration because it is purely observational and does not require registration.

**Informed Consent Statement:** All the participants gave written informed consent for participation in the study.

**Data Availability Statement:** The experimental data are available on request due to Institutional internal regulations restricting access to personal data.

**Acknowledgments:** The authors would like to thank the Irkutsk Supercomputer Center of SB RAS for providing access to the HPC-cluster «Akademik V.M. Matrosov» [44].

**Conflicts of Interest:** The authors declare no conflict of interest. The funders had no role in the design of the study; in the collection, analyses, or interpretation of data; in the writing of the manuscript, or in the decision to publish the results.

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
