# Peer review of "The Estimation of the Potential for Using Smart-Trackers as a Part of a Medical Indoor-Positioning System"

_electronics, doi:10.3390/electronics11010107_

Round 1
Reviewer 1 Report
This manuscript developed a medicine indoor-positioning system to locate the positions of patients, as well as to monitor their health statuses. I have some concerns about this system here and suggest the authors address these problems before being accepted for publication.
- The research shows the position errors of a patient in a static position (1.46 m) and in a dynamic status (3 m) in a 745x680 cm2 room. Believe it or not, this is a huge discrepancy. How will this be if tested in a 7450x6800 cm2 room? Should it be still 3 m or as large as 30 m? My advice is to perform another experiment and clear this problem. Otherwise, it’s hard to judge the real function of this system.
- In line 21, “The measurement error of pulse, saturation and diastolic pressure is less than 7%, and it is 19% for the systolic pressure.” There is no need to emphasize the accuracy of these data. Because these data are collected from the bracelet. As long as the system can receive the correct data from the token, then throw the problem to the bracelet. However, if the server received 119 while the bracelet sent 100, that is what the author should solve.
- In the “introduction” part, from line 59 to line 114, the authors are restating what the others have done. This is not the goal of an “introduction”. The authors should find the flaws in the previous research and emphasis the importance of their own research.
- In line 188, one frequency should be 2.424 GHz, instead of 2,426 GHz.
- In the “discussion” part, line 329, “Based on the research results, it can be concluded that the accuracy of positioning is insufficient at the current stage of IPS development.” However, in the “conclusion” part, line 350, “The experimental results showed that the smart-bracelets have acceptable measuring accuracy to the exclusion of systolic pressure.” They are contradictory.
Author Response
Dear reviewer, thank you very much for your interest in our work and for the clarifications made. Below are the explanations for the notes taken into account.
Point 1: The research shows the position errors of a patient in a static position (1.46 m) and in a dynamic status (3 m) in a 745x680 cm2 room. Believe it or not, this is a huge discrepancy. How will this be if tested in a 7450x6800 cm2 room? Should it be still 3 m or as large as 30 m? My advice is to perform another experiment and clear this problem. Otherwise, it’s hard to judge the real function of this system. 

Response 1: It is necessary to increase the amount of the anchors in such a way as to separate the room into coverage areas of the anchors' group when using IPS in a room with a larger size. At the same time, each group should include no less than three anchors to execute trilateration successfully and decrease the positioning error. The clarifying information was added into sections 2.4.2, 3.1, and 4.
Point 2: “The measurement error of pulse, saturation and diastolic pressure is less than 7%, and it is 19% for the systolic pressure.” There is no need to emphasize the accuracy of these data. Because these data are collected from the bracelet. As long as the system can receive the correct data from the token, then throw the problem to the bracelet. However, if the server received 119 while the bracelet sent 100, that is what the author should solve.
Response 2: We modified the “abstract” part and said that the measurements of saturation, pulse and blood pressure transmitted by the smart bracelets are similar to corresponding values received by the IPS.
Point 3: In the “introduction” part, from line 59 to line 114, the authors are restating what the others have done. This is not the goal of an “introduction”. The authors should find the flaws in the previous research and emphasis the importance of their own research.
Response 3: Most of the reviewed systems aim to locate objects, but there is no information about their ability to monitor vital signs remotely. Also, most of the reviewed systems use their own solutions as wearable hardware, which is not universal. We add this information about “introduction” part.
Point 4: In line 188, one frequency should be 2.424 GHz, instead of 2,426 GHz.
Response 4: Thank you for highlighting this point. We corrected this mistake.
Point 5: In the “discussion” part, line 329, “Based on the research results, it can be concluded that the accuracy of positioning is insufficient at the current stage of IPS development.” However, in the “conclusion” part, line 350, “The experimental results showed that the smart-bracelets have acceptable measuring accuracy to the exclusion of systolic pressure.” They are contradictory.
Response 5: This phrase is meant that the smart bracelets have acceptable measuring accuracy of the vital signs, but the accuracy of locating patients' positions is insufficient. Therefore they can be used for patients' health monitoring, but the positioning algorithm should be improved. The corresponding explicitation was added to the "discussion" part.

Reviewer 2 Report
- please add more details of the indoor position systems.
- what the accuracy of the IPS.
Author Response
Dear reviewer, thank you very much for your interest in our work and for the clarifications made. Below are the explanations for the notes taken into account.
Point 1: Please add more details of the indoor position systems. 

Response 1: We are very grateful to the reviewer for this valued suggestion. We added more information about IPS in the "introduction" part.
Point 2: What the accuracy of the IPS.
Response 2: The accuracy of the IPS in a static position is 1.46 m, and it exceeds 3 m in a dynamic. This information was added to “abstract” part.

Round 2
Reviewer 1 Report
The authors have made big progress to the last version and properly responded to the reviewers' comments. Hereby, I suggest accepting and publishing the current version as it is.